# EU27 Higher Education Institutions and COVID-19, Year 2020

**DOI:** 10.3390/ijerph18115963

**Published:** 2021-06-02

**Authors:** Sónia Rolland Sobral, Natacha Jesus-Silva, Abílio Cardoso, Fernando Moreira

**Affiliations:** 1REMIT (Research on Economics, Management and Information Technologies), Universidade Portucalense, 4200 Porto, Portugal; abilioc@upt.pt (A.C.); fmoreira@upt.pt (F.M.); 2IJP (Instituto Jurídico Portucalense), Universidade Portucalense, 4200 Porto, Portugal; natachac@uportu.pt; 3IEETA (Instituto de Engenharia Electrónica e Telemática de Aveiro), Universidade de Aveiro, 3810 Aveiro, Portugal

**Keywords:** COVID-19 pandemic, UE27, higher education institutions

## Abstract

COVID-19 forced higher education institutions to reinvent themselves. The (usually) face-to-face education has swapped to distance contingency education. This change brought about numerous challenges that impose adjustments in several dimensions, such as pedagogical strategies and the dependence on teaching platforms and computer systems—and, above all, the new relationship between the various actors (students, teachers, and management staff). All the sudden changes, combined with uncertainty concerning what was happening, created several strategies and options. This paper has the main purpose of analyzing the scientific production on higher education of EU27 academic institutions during the 2020 COVID-19 pandemic in journals indexed in Clarivate Analytics’ Web of Science and Elsevier’s Scopus. The sample is composed of 22 articles in total. The results show that the articles were published in 19 journals; their main focuses are Higher Education, COVID-19, and distance learning. In our database, we find several types of concerns, which shows that HEIs have a wide range of dimensions. We intend this article to be an instrument, not only to identify what was done in 2020, but to point out clues for the future.

## 1. Introduction

On 11 March 2020, the World Health Organization (WHO) director defined COVID-19 as a pandemic [1]. Higher education institutions (HEI) worldwide quickly had to adapt and find solutions for emergency remote education [2]. With the purpose of finding answers as quickly as possible to this problem, the HEI, at most, only had concerns about technical issues (platforms, Internet bandwidth, and so on) and “forgot” to adapt the pedagogical methodologies that characterize distance learning, even if b-learning or mixed teaching were already present in most institutions. Distance learning is characterized as planned teaching in which there is physical and/or temporal distance from the various actors in the process, using synchronous and/or asynchronous solutions [3]. Remote emergency teaching differs from traditional distance learning because it is not previously planned teaching but rather a solution to overcome the impossibility of students and teachers being physically close in the classroom [4]. Distance learning mediated by the internet can have several designations, namely its more general version e-Learning (electronic) [5], b-learning (mix of classroom and distance learning) [6], m-learning in the mobile version [7], or x-learning [8].

Abruptly, as mentioned before, all doubts and problems had to be left behind: there was a need to work on the “new normal”, continuing teaching activities. The COVID-19 pandemic forced a change in the educational process resulting in a real psychological impact for both students [9,10] and teachers [11] who had to adapt more quickly to this type of education [12]. 

The year 2020 will go down in history for several reasons—many of them quite sad. Concerning HEIs, it is essential to know what has been done and the different perspectives of stakeholders. During the pandemic crisis, several studies were carried out using various techniques such as a questionnaire on teaching and learning experiences (ETLQ) [13] or DigCompEdu Check-In questionnaire to investigate the level of digital teaching competence of Higher Education teachers [14]. Still, these studies were limited to a country or an institution [15]. Our choice was to study the 27 countries that currently belong to the European Union common project. The same type of work was done in other countries and continents geographically distant but with the same objective [16,17].

This article begins with a background section in which we make a short approach to the topic of distance learning, followed by the evolution of COVID-19 in the 27 countries of the European Union in 2020; afterwards, we present a section on the methodology of extracting scientific publications that publish about COVID-19 in EU27 institutions, presentation of results, discussion of results, and conclusions.

## 2. Background

### 2.1. Distance Learning

In the last 25 years, different experiments have been carried out. Different synchronous and asynchronous teaching objectives have been carried out, some of them with great success while others have proved to be a huge failure. It was perceived that the same strategy could be adapted to one type of student and subject, but it could not be adapted to others [18]. Teachers must know the different techniques so that they can adjust their methodologies to their audience. Effective distance learning is much more than the simple transposition of the main classes to its digital version, much more than a deposit of files on the platform adopted by the institution. Over time, several experiments were made using other pedagogical models, such as flipped classroom [19] or working in pairs [20], and using various digital media available such as portfolios, wikis, or digital tests [21]. Many studies reveal that some teachers appeared to be well prepared for a situation, so, if similar online teaching scenarios were ever repeated, the quality of teaching seems to be guaranteed [22]. A study establishes five priority attributes of online teaching to achieve quality: interaction between students, level of student concentration in online classes, improvement of the online test review system, and improved student satisfaction with online teaching and diversity of assessment tests [23].

### 2.2. COVID-19 and EU27

The 27 countries that currently make up the European Union have very diverse geography and population density. Germany has the largest population (83,992,315), and Malta has the smallest population (442.46). On average, each country has 16.498.22 inhabitants [24]. In 2020, there was a first wave of the COVID-19 pandemic between February and May, and another wave started in September. The following figure shows the cases and deaths for each month of the year 2020 in cumulative terms for the 27 countries of the European Union. This last wave started in July for Spain and France.

The following figure (Figure 1) shows the monthly data COVID-19 cases and deaths for the UE27 countries [25].

The following figure (Figure 2) shows the monthly data COVID-19 cases for each of the 27 countries [25].

The following figure (Figure 3) shows the monthly death COVID-19 data for each of the 27 countries.

Luxembourg, Czechia, and Belgium are the three countries that registered the most COVID-19 cases in 2020 considering the population of their country. Belgium, Italy, and Spain are the three counties that reported the most COVID-19 deaths in 2020 considering the people of their country.

## 3. Bibliometric and Systematic Literature Review

Bibliometrics refers to the research methodology employed in library and information sciences, which utilizes quantitative analysis and statistics to describe articles’ distribution patterns within a given topic, field, institution, or country [26]. Bibliometrics is usually used for the quantitative research assessment of academic output, and it is starting to be used for practice-based research [27]. A bibliometric analysis of international papers is a way to provide a valuable reference for future research [28]. A systematic literature review is a method to analyze, evaluate, and interpret each study relevant to a particular research question, specific area, or phenomenon of interest [29]. This process was originated in medical science due to the increasing amount of research in each area [30]. Consequently, it was necessary to identify and guide the research towards an uninvestigated subject [31]. The scientific community has proposed steps to apply these protocols, more specifically in the software engineering area.

Kitchenham and Charters [30] presented a set of guidelines for planning, conducting, and reporting a systematic review: developing a protocol, defining the research question, specifying what will be done to address the problem of a single researcher, applying inclusion/exclusion criteria and undertaking all the data extraction, defining the search strategy, defining the data to be extracted from each primary study including quality data, maintaining lists of included and excluded studies, using the data synthesis guidelines, and using the reporting guidelines. Brereton et al. [32] propose ten steps for adopting systematic review within the software engineering domain: specify research question, develop review protocol, validate review protocol, identify relevant research, select primary studies, assess study quality, extract required data, synthesize data, write review report, and validate report. 

The process of this methodology is presented in the following sections. In this work, we adapted the steps defined by previous studies and created a method framed in our research. The main objective of this bibliometric and systematic literature review was to obtain essential data on scientific production considering the approach by academics and researchers in journal documents to identify what the scientific production of researchers with affiliation in the 27 European countries about higher education was during the COVID-19 pandemic of the year 2020. 

For this purpose, it was planned to search the different databases for relevant papers, and we consider that the following questions are essential for the investigation:RQ1: Where were the articles published?RQ2: What is the focus of the articles? RQ3: Who publishes on the subject? Where do researchers work?RQ4: What are the purpose, approach, and findings of the articles?

### 3.1. Search Strategy

We have built a query string, and logical operators complement this to improve the execution results. We limited the search process to documents that had been published only in journals. For each database, it was necessary to build a specific query because each one has a different syntax; and an example of a resulting query is shown below.

The Scopus search strategy was: TITLE-ABS-KEY ((‘covid’) OR (‘COVID’) OR (‘Covid-19’) OR (‘COVID-19’)) AND ((‘Higher education’) OR (‘University’)) AND (‘survey’).DocType: ArticleAffiliation Country E27

The WoS search strategy was: TS = ((‘covid’) OR (‘COVID’) OR (‘Covid-19’) OR (‘COVID-19’)) AND ((‘Higher education’) OR (‘University’)) AND (‘survey’)Document Type (ARTICLE)AD (Country E27)

### 3.2. Exclusion Criteria

Papers Discarded by the Abstract

Papers Discarded by Full Text

### 3.3. Inclusion Criteria

Papers Included in the Databases of Table 1

Papers as a Result of Journals

Higher Education’s Studies

Countries from E27

### 3.4. Collected Information

We consider different databases to execute the search strings. Access to databases is private; the databases are shown in Table 1.

## 4. Bibliometric Results

A set of 34 published papers were collected from WoS and 39 from Scopus. The search returned a total of 52 articles and reviews after discounting the duplicate results. After applying the criteria that constitute mandatory eligibility requirements (first by title and anstract, then by full-text), the number of papers is now 22: seven Scopus, nine WoS, six in both databases. Our goal was to study what happened in 2020; however, eight articles are published at the beginning of 2021. A PRISMA [33] flow chart of the selection process and screening is provided in Figure 4.

The articles were published in 19 journals: three articles in Sustainability, two articles in the International Journal of Environmental Research and Public Health. The HIndex of the articles is diverse: from 1 to 95. The same happens with the SJR: from 0 to 0.91. There are 33 different sub-categories: the most frequent is Social Sciences (Sociology and Political Science) with three occurrences. In addition, 38% belong to the second quartile, and 27% belong to the first quartile. Table 2 shows all publications where the papers for this study are found: title of the source (source), number of papers in our study (N), HIndex, Scientific Journal Rankings (SJR,) category and subject, as well as the quartile associated with each sub-category (Q).

The articles have 90 different keywords, and only nine occur at least twice: Higher education (12), COVID-19 (9), distance learning (5), COVID-19 pandemic (4), E-learning (3), Pandemic (3), Coronavirus (2), and online learning (2). Figure 5 shows the cloud of keywords provided by the authors, using wordclouds.com facilities (accessed 27 march 2021).

Five of the papers were written by three authors and six articles by four authors. There are also articles with one, two, and three co-authors, an article with eleven authors, and another one with 17 authors. There are 94 different authors. Two of the authors present two papers: Mariia Rizun and Artur Strzelecki, both from the University of Economics in Katowice, Katowice, Poland. One article is written by both, the other article being written with two colleagues: Karina Cicha and Paulina Rutecka.

There are 65 different affiliation addresses, eight of which appear at least twice:Univ Granada, Fac Educ Sci, Dept Didact and Sch Org, Granada, SpainUniversity of Zielona Gora Univ Zielona Gora, Zielona Gora, PolandDepartment of Informatics, University of Economics in Katowice, Katowice, 40–287, PolandEotvos Lorand Univ, Fac Primary and Presch Educ, Budapest, HungaryNúcleo Migrare, Centro de Estudos Geográficos, Instituto de Geografia e Ordenamento do Território, Universidade de Lisboa, Lisboa, PortugalTallinn Univ, Sch Educ Sci, Narva Mnt 25, EE-10120 Tallinn, EstoniaUniv Catolica Portuguesa, Catolica Porto Business Sch, Porto, PortugalUniv Granada, Dept Didact & Sch Org, Fac Educ Econ and Technol, Ceuta, Spain.

Nineteen countries appear as the authors’ affiliation address, Spain (26%) and Portugal (23%) are the most frequent. Figure 6 shows the percentage of each country in the database of our study.

Authors generally publish articles with affiliations from one country. There are three exceptions:(a)France and Algeria(b)Spain, Ecuador, and Italy(c)Germany, United Kingdom, Malta, Brazil, Portugal, and South Africa.

## 5. Systematic Literature Review

The studies found concern eleven of the 27 countries of the European Union. Spain and Poland with 23% each, Portugal with 14%, Romania with 9% and Cyprus, Czech Republic, Estonia, France, Hungary, Latvia, and Italy with 4.5%.

The focus of most articles is distance learning or the change from face-to-face to distance learning. There are several perspectives, like mobile learning methodology [34] and gamification [35], for a specific subject [36,37]; for a specific group of students, such as international mobility students [38] or first-year students [39]; concerned with the quality of educational process on online platforms [40], the level of communication of social responsibility by higher education institutions [41] or technical conditions of distance learning [42]; adaptation of the learning process [43], concern about connecting to the digital world [44,45,46]; the influence of some factors on students’ acceptance of shifting education to distance learning [47], Evaluation of the Emotional and Cognitive Regulation [48], comparison of face-to-face classes with distance learning [49] by the student’s perspective [50,51], teacher’s perspective [43,52], organization [53], or various actors [54].

Almost all papers included a survey in their methodology. We found surveys to be answered only by teachers, with samples between 100 and 200 [36,43,52] or with a size of 1544 respondents [34]. There are studies with surveys to the entire academic community [53]. Most studies included surveys answered by students. the sample size varies widely: from 100 or less [38,39,51], 250 or less [37,40,45,46,49,50,55], less than 1000 students [42,44,54], or more than 1000 [47,48]. Two studies use the General Extended Technology Acceptance Model for E-Learning (GETAMEL) [39,47], one study uses the Digital Citizenship Behavior Scale (DCB) [45] and another uses the Cognitive Emotion Regulation Questionnaire (CERQ) [48]. One of the studies uses three surveys: at the beginning, in the middle, and at the end of the semester [35], while another compares student surveys in the semester prior to the pandemic with the surveys of students who participated in the course during the pandemic [49]. Only two papers do not have surveys: one acquires information through the content of the websites of public institutions of higher education [41], while the other monitors students and their digital footprint [51].

There are many different results, just as would be expected and will be presented in the following paragraphs:

The results do not show statistically significant group differences in the students’ intrinsic and extrinsic motivation, feelings of attachment to the university, and absorption and vigor involvement dimensions. However, a moderate negative effect was found in the dimension of engagement and dedication [49].

The digital divide still exists between the most developed countries in Nordic Europe and the least developed countries in southern and eastern Europe. Still, Latvian higher education institutions have significantly increased digital content in external and internal communication systems. They can offer competitive educational services [44].

Difficulties in adapting to new methods of distance education, housing deficit, and mental health problems were identified, especially in women [38].

We also underline the importance of parents’ education since it is the families with a high or medium-high level of education who have a computer for exclusive use and its high-speed connectivity for accurate monitoring of virtual teaching, and also the lack of teachers’ adaptation to the personal circumstances of students [54].

The results of the research indicated that the most important factors that influence the feelings of students and can convince them to change from teaching in the classroom to teaching in the distance learning model are the feeling of pleasure in this form of education and a sense of self-efficacy [39].

Students affirmed their total satisfaction with the learning process, the use of distance tools, and the level of mastery of these tools by their teachers. The results of knowledge tests show that, for the same course, distance learning does not reduce engineering students’ performance. They obtained local grades like those expected in face-to-face education [51].

Students have a medium feeling that distance learning has been enhancing their effectiveness and productivity. Their self-efficacy with distance learning is also medium; students consider distance learning IT tools to be very intuitive, and they are generally comfortable using computers and the internet; they plan to use distance learning often during the semester. However, despite the positive opinions about distance education, the students would like to go back to traditional education [47].

The flexibility of schedules, the convenience of learning at home, and the opportunity for self-development are favorable points for distance learning [42]. However, the time devoted to studying has not increased [48].

Problems with personal relationships and communication, self-regulated learning, and technical problems are shown [42]. The negative anxiety of students due to the pandemic is reflected in their e-learning processes [45]. A negative correlation was also reported between the impact of the pandemic on the academic path and the attitudes of Portuguese students in higher education concerning the mandatory nature of the digital university and distance learning [55].

Students were more satisfied with the online classes the more positive their attitudes towards the impact of the pandemic, the level of HEI preparation and adaptation, satisfaction with the evaluation format, and the overall experience balance [55].

Students have shown preference for synchronous learning, sometimes combined with asynchronous support [43]. Teachers and students show their preference for being present, but they recognize the justification for the change of scenery and identify positive elements in virtuality [53].

The students’ most used equipment was the smartphone, and 42% attended online courses on a smartphone [46]. Situations were found where few students had the necessary infrastructure to guarantee the smooth running of the teaching/learning process, namely in Romania, and, in their cases, communication with teachers was ineffective. Many students report a lack of motivation for learning. They also report problems with inadequate online assessment, lack of socialization, and interaction, potentially affecting mental and physical health also caused by physical inactivity [40].

The level of social responsibility communication by higher education institutions in the Czech Republic is low [41]. The sampled HEI have adequate infrastructure to continue to teach during the lockdowns [37]. The institution’s management needs to invest in additional infrastructure and educational technologists who would support the teaching staff to systematically widen their pedagogical repertoire and raise their digital competencies to the next level [52]. Some recommendations are given to organizations: good communication, providing information about the change; involve students in making decisions related to the transformations carried out; adjust the content and teaching method to the way of online learning; take care of social presence using synchronous forms; limit the tools used, preferably to choose one; provide support in the field of technologies used, which enable participation in online learning (such as: facilitating access to libraries); and pay special attention to first-year students [50].

## 6. Discussion and Conclusions

Our main objective with this article is to know the concern and what was investigated about the EU27 HEI during the 2020 pandemic phase. We found two waves, or peak number of COVID-19 cases, and a consequent lockdown in most countries. This confinement led to the closure of the facilities that continued to operate at a distance—what became known as remote emergency teaching. We believe that the year 2020—and from what we are seeing, also the year 2021—is a year that remains in history for several reasons—and everything that happened, how the associations reacted, and how the various actors acted must be described. All of this information will be very useful in the future.

First of all, we have to answer our research questions:RQ1: Where were the articles published?

The articles were published in 19 journals: three articles in Sustainability, two articles in the International Journal of Environmental Research and Public Health, both published by MDPI.

RQ2: What is the focus of the articles?

The articles have 90 different keywords, and only nine occur at least twice: The most used keywords are higher education (12), COVID-19 (9) and distance learning (5).

RQ3: Who publishes on the subject? Where do researchers work?

There are 94 different authors. Two of the authors present two papers: Mariia Rizun and Artur Strzelecki, both from the University of Economics in Katowice, Katowice, Poland.

RQ4: What are the purpose, approach, and findings of the articles?

The focus of most articles is distance learning. Almost all papers included a survey in their methodology. There are many different results: some show that students liked online classes and others did not. There are equipment problems and a lack of pedagogical knowledge to transform traditional teaching into distance learning. There are recommendations for organizations.

This paper elaborated a study of the scientific publication in Journals on higher education institutions in pandemic times in the 27 countries of the European Union. It seemed to the authors that it would be essential to limit the study to countries that belong to the European community. In terms of the health crisis, we saw two waves: February until May and another wave that for most countries started in September and continued until the following year. These waves—an increase in COVID-19 cases and resulting deaths—forced countries to establish lockdowns and, consequently, they had to close universities. This situation meant that institutions were forced to reinvent themselves, rethinking their processes. Classes moved to teach on platforms other than the classroom context—usually called contingency teaching because it is unplanned distance learning. Teachers had to change their pedagogical strategies and get used to not teaching in the traditional classroom. The services of the institutions had to continue working by remote work without prejudice to the proper functioning. The organizations had to make decisions following their country’s directives and tried to minimize the effects of the lockdown.

Our article summarizes what was done in 2020, that is, what were (in that terrible year) the concerns of the HEIs in the 27 EU countries. We realized that there are many dimensions of concerns: there are some issues that are already being worked on and which have very important conclusions. Thus, it is possible to draw said conclusions and take advantage of the work of other researchers so as not to return to work on the same subjects to reach the same conclusions priorly mentioned.

This study focused on the year 2020. Unfortunately, the year 2021 continues to be a year marked by COVID-19. As future work, we would like to assess what EU27 HEIs have learned concerning the 2020 experience.

The year 2020 will go down in history because of the COVID-19 pandemic crisis and the need for the “new normal” to cause as minor damage as possible—even taking the opportunity to weigh and improve many institutions sectors. We have to think about whether the Europe of 27 was prepared, how it prepared, and how it is currently prepared for a new crisis—be it a pandemic crisis or any other that we cannot predict.

## Figures and Tables

**Figure 1 ijerph-18-05963-f001:**
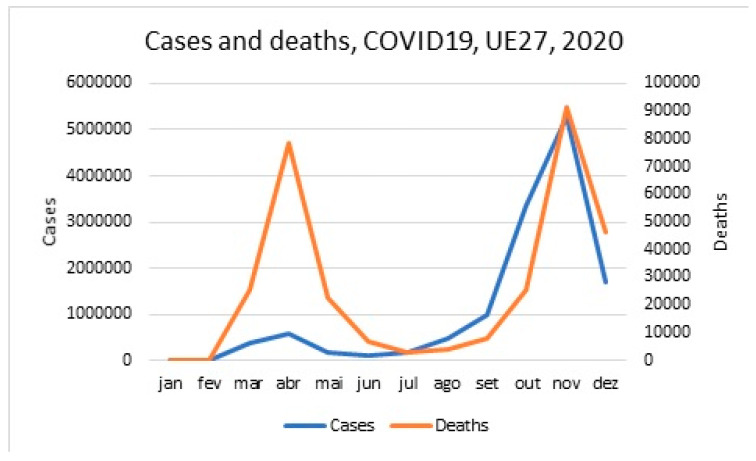
COVID-19 cases and deaths, 2020, UE27.

**Figure 2 ijerph-18-05963-f002:**
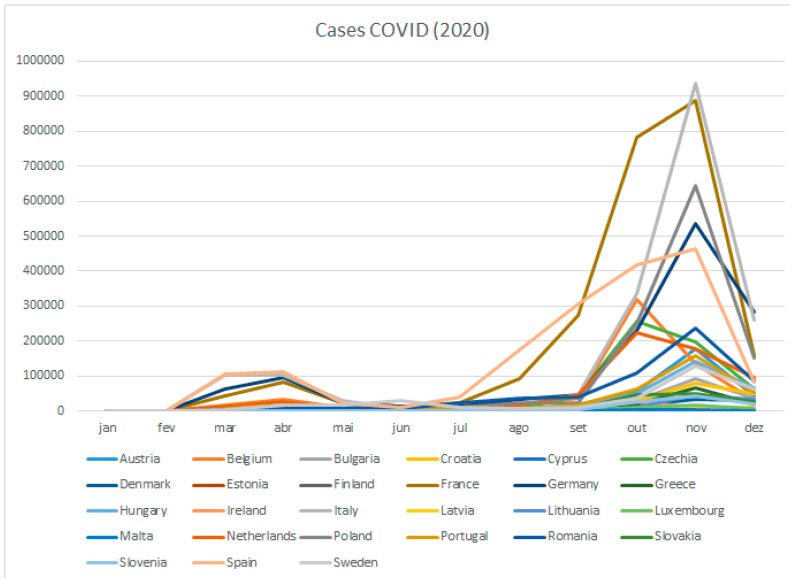
COVID-19 cases Eu27, 2020.

**Figure 3 ijerph-18-05963-f003:**
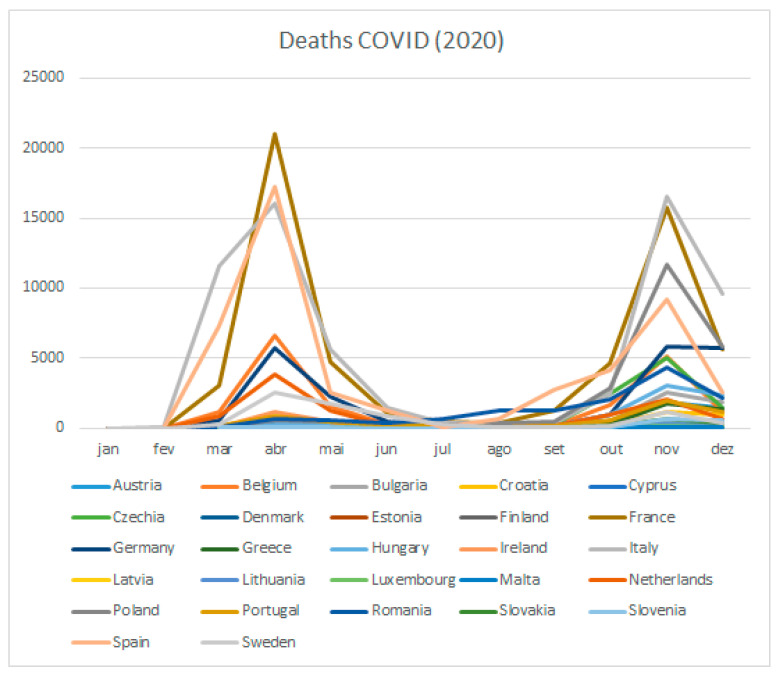
COVID-19 deaths, 2020 by country.

**Figure 4 ijerph-18-05963-f004:**
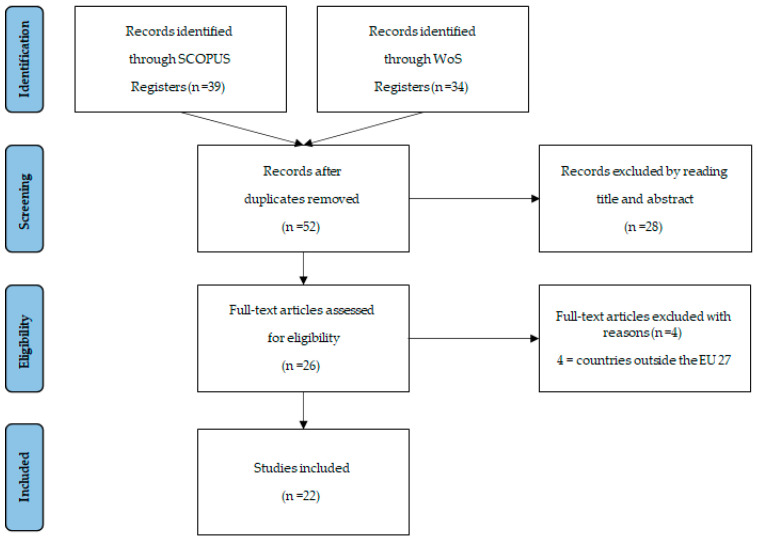
PRISMA flowchart of the systematic literature review.

**Figure 5 ijerph-18-05963-f005:**
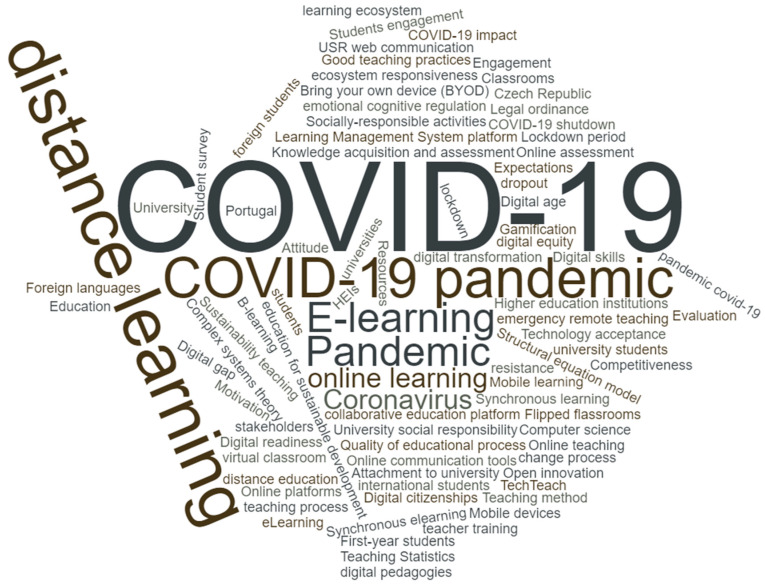
Cloud of keywords provided by the authors.

**Figure 6 ijerph-18-05963-f006:**
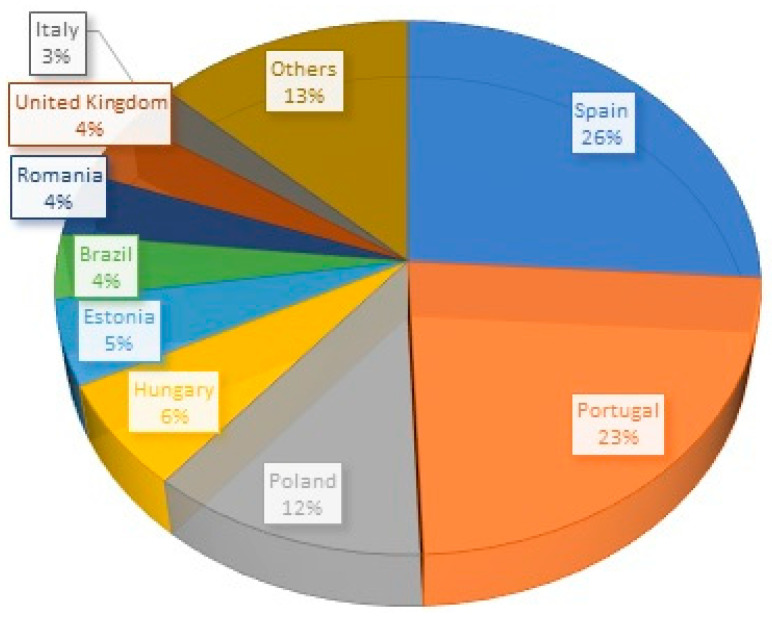
Countries of affiliation of the authors.

**Table 1 ijerph-18-05963-t001:** Databases used in the search (accessed on 26 February 2021).

Name	Acronym	URL
Scopus	Scopus	https://www.scopus.com/
Web of Science	WoS	https://apps.webofknowledge.com/

**Table 2 ijerph-18-05963-t002:** Journal information.

Source	N	Hindex	SJR	Subject and Category	Q
Accounting Research Journal	1	13	0.32	Business, Management and Accounting (Accounting)	Q3;
				Economics, Econometrics and Finance (Finance)	Q3
Boletin de Estadistica e Investigacion Operativa	1	3	0.12	Decision Sciences (Management Science and Operations Research);	Q4;
				Decision Sciences (Statistics, Probability and Uncertainty)	Q4
Civil Szemle	1	1	0	Social Sciences (Public Administration);	Q4;
				Social Sciences (Sociology and Political Science)	Q4
Contemporary Educational Technology	1	4	0.43	Business, Management and Accounting (Management of Technology and Innovation);	Q2;
				Social Sciences (Education)	Q2
Education Sciences	1	7	0.24	Psychology (Developmental and Educational Psychology);	Q4;
				Social Sciences (Education);	Q3;
				Social Sciences (Public Administration)	Q3
E-mentor	1	*		Education & Educational Research	
Emerging Science Journal	1	New			
Environment, Development and Sustainability	1	52	0.55	Economics, Econometrics and Finance (Economics and Econometrics);	Q2;
				Environmental Science (Management, Monitoring, Policy, and Law);	Q2;
				Social Sciences (Geography, Planning and Development)	Q2
FINISTERRA-REVISTA PORTUGUESA DE GEOGRAFIA	1	*		Geography	
Frontiers in Psychology	1	95	0.91	Psychology (Psychology (miscellaneous))	Q1
IEEE Access	1	86	0.78	Computer Science (Computer Science (miscellaneous));	Q1;
				Engineering (Engineering (miscellaneous));	Q1;
				Materials Science (Materials Science (miscellaneous))	Q2
Information	1	20	0.35	Computer Science (Information Systems)	Q3
INTERACTION DESIGN AND ARCHITECTURES	1	*		Education & Educational Research	
International Journal of Engineering Pedagogy	1	3	0.24	Engineering (Engineering (miscellaneous));	Q2;
				Social Sciences (Education)	Q3
International Journal of Environmental Research and Public Health	2	92	0.74	Environmental Science (Health, Toxicology and Mutagenesis Pollution);	Q2
				Medicine (Public Health, Environmental and Occupational Health)	Q2
Journal of Open Innovation: Technology, Market, and Complexity	1	20	0.78	Economics, Econometrics and Finance (Economics, Econometrics and Finance (miscellaneous));	Q1; Q1; Q1
				Social Sciences (Development);	
				Social Sciences (Sociology and Political Science)	
RIED-REVISTA IBEROAMERICANA DE EDUCACION A DISTANCIA	1	*		Education and Educational Research	
Sustainability	3	68	0.58	Energy (Energy Engineering and Power Technology); Energy (Renewable Energy, Sustainability and the Environment);	Q2; Q2; Q2
				Environmental Science (Environmental Science (miscellaneous)); Environmental Science (Management, Monitoring, Policy, and Law);	
				Social Sciences (Geography, Planning, and Development)	
XLinguae	1	14	0.32	Arts and Humanities (Language and Linguistics); Arts and Humanities (Philosophy);	Q1; Q1; Q1
				Social Sciences (Linguistics and Language)	

*: No SCOPUS Hindex.

## Data Availability

The data presented in this study are available on request from the corresponding author.

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
