# Peer review of "EU27 Higher Education Institutions and COVID-19, Year 2020"

_ijerph, 2021, doi:10.3390/ijerph18115963_

Round 1

Reviewer 1 Report

This is an interesting manuscript that (in my opinion) could enrich the scientific discourse regarding the impact of the Covid-19 pandemic. However, in my opinion, the manuscript needs a major revision due to the following reasons:

  • I have noticed some typographical errors in the manuscript. Therefore, the authors should get the manuscript professionally proofread before resubmission.
  • My main problem with this manuscript is that it does not reveal the authors' epistemological interest. Even the research questions do not provide meaningful information on this. Therefore, the authors should fundamentally revise their introduction and research questions regarding their epistemological interest.
  • I also do not understand the need for chapter 2.2. In my opinion, this chapter might be omitted entirely.
  • Furthermore, I miss a critical analysis of the literature reviewed by the authors. What is to be criticized about the existing studies? What is missing? What should future research address? Unfortunately, I do not get any answers to these questions.

Author Response

This is an interesting manuscript that (in my opinion) could enrich the scientific discourse regarding the impact of the Covid-19 pandemic. However, in my opinion, the manuscript needs a major revision due to the following reasons:

Thank you very much for your comments: the suggestions were very important to improve the document that we now send you.

We chose to view each of the improvement suggestions and respond one by one.

In yellow are the suggestions and below each suggestion is our commentary-explanation.

I have noticed some typographical errors in the manuscript. Therefore, the authors should get the manuscript professionally proofread before resubmission.

Each of us reread the document and found some problems that we have already solved.

My main problem with this manuscript is that it does not reveal the authors' epistemological interest. Even the research questions do not provide meaningful information on this. Therefore, the authors should fundamentally revise their introduction and research questions regarding their epistemological interest.

The interest in these issues is to know what was done in 2020 - not only to report what was done, but also to learn lessons for the future. We incorporate this indication both in the introduction and in the conclusions. We think that our objective has now been clarified. (line 20, 21, 46-54, 132-137, 319-321, 376-380)

I also do not understand the need for chapter 2.2. In my opinion, this chapter might be omitted entirely.

We removed the whole chapter 2.2,

Furthermore, I miss a critical analysis of the literature reviewed by the authors. What is to be criticized about the existing studies? What is missing? What should future research address? Unfortunately, I do not get any answers to these questions. Our goal was to portray what exists in 2020 to be able to learn lessons for the future: not to rediscover the wheel. With our article you can see how the EU27 HEIs mobilized to respond to what happened (lines 367-375).

Thank you very much for the suggestions and for the time you spent with us. We hope that the improved article is in line with your expectations.

Reviewer 2 Report

I do appreciate the opportunity to review the manuscript COVID in higher education EU27 institutions. The paper elaborates on an important issue of switching to remote learning due to COVID 19 pandemic. This topic should be developed by the scientific community. The authors identified vital papers in the matter.

Generally, I would advise including a systematic approach already proven viable into research- like PRISMA. Please see details http://prisma-statement.org/PRISMAStatement/FlowDiagram.

The tool has been verified and checked, therefore using it in scientific papers seems to be the right move.

I would also like to see a broader range of databases. At least Google Scholar should be included as a freely available resource in those databases.

I would advise including legend to the tables, e.g., what the N stands for, SJR.

I would like to see some images/print screens presenting systems described in the section concerning the exemplary university.

Also, the exclusion criteria require to be reformulated, e.g., Papers discarded by name

I do not have information on what was incorrect in the name to discard the paper.

Author Response

I do appreciate the opportunity to review the manuscript COVID in higher education EU27 institutions. The paper elaborates on an important issue of switching to remote learning due to COVID 19 pandemic. This topic should be developed by the scientific community. The authors identified vital papers in the matter.

Thank you very much for your comments: the suggestions were very important to improve the document that we now send you.

We chose to view each of the improvement suggestions and respond one by one.

In yellow are the suggestions and below each suggestion is our commentary-explanation.

Generally, I would advise including a systematic approach already proven viable into research- like PRISMA. Please see details http://prisma-statement.org/PRISMAStatement/FlowDiagram. The tool has been verified and checked, therefore using it in scientific papers seems to be the right move.

Thank you very much for the suggestion. We know PRIMA but we chose to use a very similar approach.

I would also like to see a broader range of databases. At least Google Scholar should be included as a freely available resource in those databases.

Thank you very much for the suggestion. Our opinion was to limit the study to WoS and SCOPUS - we could have used GoogleScholar (or another database), but we chose to use the two databases that deserve more confidence.

I would advise including legend to the tables, e.g., what the N stand a freely available resource in those databases.s for, SJR.

Change made. Thanks (line 182-190).

I would like to see some images/print screens presenting systems described in the section concerning the exemplary university.

We removed the whole chapter 2.2 because it was suggested by one of the reviewers.

Also, the exclusion criteria require to be reformulated, e.g., Papers discarded by name

I do not have information on what was incorrect in the name to discard the paper.

Change made. Thanks (Line 160).

Thank you very much for the suggestions and for the time you spent with us. We hope that the improved article is in line with your expectations. 

Reviewer 3 Report

Interesting and actually research. I suggest some modifications:

CONTENT

  • How important is the epidemiological impact of the different EU countries in this article? it should be important how the confinement has impacted HEI
  • I think that the discussion section reflects more a summary of results than a cross-examination of results and status of the issue. Too many arguments are repeated.
  • The conclusions are generic. Better to focus them on the implications in HEI

FORM

  • Check how the figures are quoted in the text (figure 1 and 2)
  • Review content figure 2 (27 countries?)

Author Response

Interesting and actually research. I suggest some modifications:

Thank you very much for your comments: the suggestions were very important to improve the document that we now send you.

We chose to view each of the improvement suggestions and respond one by one.

In yellow are the suggestions and below each suggestion is our commentary-explanation.

CONTENT

How important is the epidemiological impact of the different EU countries in this article? it should be important how the confinement has impacted HEI

We made changes to the document so that those who read it could better understand what we wanted. We think that our goal is already clarified in this enhanced version.

I think that the discussion section reflects more a summary of results than a cross-examination of results and status of the issue. Too many arguments are repeated.

Thank you for your suggestion. It has been revised. What we wanted to do was summarize the results, listing and analysing the different approaches. (line 313)

The conclusions are generic. Better to focus them on the implications in HEI

Thanks for the suggestion. We improve the conclusions so that the reader can better understand what our goal is. (line 387)

FORM

Check how the figures are quoted in the text (figure 1 and 2)

Change made. Thanks (line 89, 94)

Thank you very much for the suggestions and for the time you spent with us. We hope that the improved article is in line with your expectations.

Review content figure 2 (27 countries?

Change made. Thanks (line 96).

Round 2

Reviewer 1 Report

The authors have conscientiously and comprehensively integrated my comments on their manuscript (and those of the other reviewers). In my opinion, the article can therefore be accepted in its current form.

Author Response

Thank you very much for your comments: the suggestions were very important to improve the document that we now send you.

Reviewer 2 Report

Thank you for the opportunity to revisit the paper. However, most of my suggestions were not taken into account. I do not understand why the authors insist on unstandardized Figure 4, where Prisma provides proven and standardized steps in the manner of researching the literature on the topic. 
I do think that two databases for a systematic review covering such a short period of time are not enough. I do understand the value of Medline and the web of science, but I see the need to widen the search to at least 4.

Additionally, I am missing a table summarising the content of included manuscripts. There is only a description in the text ( systematic literature review section).

I understand that authors anchor their research in the engineering literature review; however, the journal they decided to submit the manuscript demands a different approach, in my opinion. Therefore, I would still recommend those changes.
Minor comments:

line 176 “After applying the criteria that constitute mandatory eligibility requirements...”- which mandatory criteria are mentioned here and who mandated them.

The authors mentioned that access to those databases is private. I would like to see an explanation of how the access was granted to the authors- institutional approval or maybe other.

Inclusion criteria enumerate higher education studies. What do authors understand by higher education? University of all types of degrees? Or maybe the first two cycles of studies? I would request further explanation. 

Next, it appears as a result of a journal as a criterion of inclusion- peer-reviewed or any journals, or with IF... please explain further.

Countries of EU- authors to be citizens of those countries or represent institutions based in those countries? Please explain further.

Author Response

Thank you for the opportunity to revisit the paper. However, most of my suggestions were not taken into account.

Thank you very much for your comments: the suggestions were very important to improve the document that we now send you.

I do not understand why the authors insist on unstandardized Figure 4, where Prisma provides proven and standardized steps in the manner of researching the literature on the topic. 

We agree with your doubts about the non-standardized Figure 4. Now, a PRISMA flowchart of the selection and screening process is provided in Figure 4.

I do think that two databases for a systematic review covering such a short period of time are not enough. I do understand the value of Medline and the web of science, but I see the need to widen the search to at least 4.

The authors have chosen only two databases for this systematic review using the following criteria: Journal Ranking Lists (Journal Citation Reports from WoS; Scimago from Scopus) and indexation features (WoS and Scopus).

Additionally, I am missing a table summarising the content of included manuscripts. There is only a description in the text ( systematic literature review section).

The synthesis of the content of the articles is on lines 346 - 350 and all information in text format is in the Systematic Review of Literature (starting on line 232). The authors made a summary table, but that table was difficult to read because the records have very different contents. For this reason, we chose to remove that summary table and keep only the text presented.

I understand that authors anchor their research in the engineering literature review; however, the journal they decided to submit the manuscript demands a different approach, in my opinion. Therefore, I would still recommend those changes.

We agree with the suggestion made; however, the investigation did not consider any area, but the aim is to understand the main concerns of higher education institutions in the EU27 in 2020 related to the lockdown of COVID-19.

Minor comments:

line 176 “After applying the criteria that constitute mandatory eligibility requirements...”- which mandatory criteria are mentioned here and who mandated them.

The criteria that constitute mandatory eligibility requirements are those indicated in lines 164-167

The authors mentioned that access to those databases is private. I would like to see an explanation of how the access was granted to the authors- institutional approval or maybe other.

Access to the WoS and Scopus databases was carried out through the institution's subscriptions to which the authors belong.

Inclusion criteria enumerate higher education studies. What do authors understand by higher education? University of all types of degrees? Or maybe the first two cycles of studies? I would request further explanation. 

The authors' understanding regarding higher education has to do with university education and all cycles of studies.

Next, it appears as a result of a journal as a criterion of inclusion- peer-reviewed or any journals, or with IF... please explain further.

It was considered only journals with peer reviewed. The IF and quartile were not considered as inclusion criteria, but we decided to include both as complementary information in table 2.

Countries of EU- authors to be citizens of those countries or represent institutions based in those countries? Please explain further.

All papers considered in this study are representing institutions based on E27 countries, and the authors could be, or not, citizens of those countries.

Thank you very much for the suggestions and for the time you spent with us. We hope that the improved article is in line with your expectations.

Reviewer 3 Report

Form:

- I suggest to review number of figure on line 196.

- I suggest to change line 324 that still mentions a disapeared section (old version)

Author Response

Thank you very much for your comments: the suggestions were very important to improve the document that we now send you.

  • I suggest to review number of figure on line 196.
    • Done
  • I suggest to change line 324 that still mentions a disapeared section (old version)
    • We deleted the entire paragraph.

Thank you very much for the suggestions and for the time you spent with us. We hope that the improved article is in line with your expectations.